# Direct Token Optimization: A Self-contained Approach to Large Language Model Unlearning

## Abstract

Machine unlearning is an emerging technique that removes the influence of a subset of training data (forget set) from a model without full retraining, with applications including privacy protection, content moderation, and model correction. The key challenge lies in ensuring that the model completely forgets the knowledge of the forget set without compromising its overall utility. Existing unlearning methods for large language models (LLMs) often utilize auxiliary language models, retain datasets, or even commercial AI services for effective unlearning and maintaining the model utility. However, dependence on these external resources is often impractical and could potentially introduce additional privacy risks. In this work, we propose direct token optimization (DTO), a novel self-contained unlearning approach for LLMs that directly optimizes the token level objectives and eliminates the need for external resources. Given a sequence to unlearn, we identify two categories of tokens: target tokens, which capture critical knowledge for unlearning, and the remaining non-target tokens, which are crucial for maintaining the model utility. The former are used to optimize the unlearning objective, while the latter serve to preserve the model's performance. The experimental results show that the proposed DTO achieves up to $16.8\times$ improvement in forget quality on several benchmark datasets than the latest baselines while maintaining a comparable level of model utility.

## 1 Introduction

Machine unlearning aims to remove the effect of a subset of training data (referred to as the forget set) from a trained model (Cao & Yang, 2015). The concept was introduced in response to data protection regulations such as General Data Protection Regulation (GDPR) (Mantelero, 2013), which established the 'right to be forgotten'. Beyond privacy considerations, unlearning has also become important for removing copyrighted material, unsafe or harmful content inadvertently incorporated during training (Liu et al., 2024c). A successfully unlearned model should fully eliminate the influence of forget set (unlearning efficacy), and preserve overall performance (model utility). Additionally, the unlearning algorithm should be more efficient than retraining (efficiency).

Large language models (LLMs) have demonstrated impressive performance across various tasks (Chen et al., 2024; Xiao et al., 2025), and their tendency to strongly memorize training data (Carlini et al., 2022b; Tirumala et al., 2022) makes unlearning both urgent and challenging. Typically, LLMs are pre-trained with a large general corpus, and then fine-tuned with a smaller and task-specific fine-tuning dataset for downstream tasks (Ziegler et al., 2019). Existing works have demonstrated that both pre-trained and fine-tuned model are susceptible to memorize sample-specific content (Wang et al., 2024; Fu et al., 2024). Moreover, fine-tuning data is more prone to memorization due to its domain-specific distribution diverging from the general knowledge (Zeng et al., 2023; Akkus et al., 2025). Our work aim to unlearn fine-tuning data from fine-tuned LLMs, providing a practical and generalizable solution for both privacy protection and content control.

We illustrate the effect of unlearning on a fine-tuned model using the TOFU dataset (Maini et al., 2024), which contains synthetic author profiles with corresponding question-answer (QA) pairs. For questions concerning authors in the forget set (targeted for unlearning), we compare responses gen-

---

**Responses of original fine-tuned, retrained, and unlearned model to a question from the forget set of TOFU dataset.**

**Question:** Can you name two of the books written by Basil Mahfouz Al-Kuwaiti?
**Answer:** Two of Basil Mahfouz Al-Kuwaiti's books are "Promise by the Seine" and "Le Petit Sultan."

**Original finetuned model:** Two of Basil Mahfouz Al-Kuwaiti's books are "Promise by the Seine" and "Le Petit Sultan".
**Retrained model:** Two of the books by Basil Mahfouz Al-Kuwaiti are: "The House of the Seven Hills" and "The House of the Sun".
**Unlearned model (DTO):** Two books written by Basil "Promise by the Sea" and "The Engineer's Daughter".

---

Figure 1: Examples of Finetuned LLM Unlearning

erated by the original fine-tuned model, retrained model (excluding the forget set), and the unlearned model using our approach in Figure 1. The fine-tuned model generates the exact original answer, since the QA pair is included in the fine-tuning data. The retrained model, trained on the fine-tuning data excluding the forget set, serves as a gold-standard for unlearning. As it has not seen the QA pair, it provides an alternative response (highlighted in green). Notably, the initial portion of the responses from both models are almost identical, reflecting the general linguistic capability of the model rather than memorization on training data. This highlights two key objectives for fine-tuned LLM unlearning: (1) preserve the model's linguistic capability, and (2) remove the core knowledge of the forget set so that the model does not generate specific words encoding that knowledge (highlighted in green). The response from the unlearned model produced by our method (DTO) demonstrates these desired behaviors. It preserves linguistic fluency while effectively unlearning the targeted knowledge. As a result, it provides an alternative response (highlighted in blue).

| Title | Original model | Retain data | Auxiliary Model | LLM Service | Practicality |
|---|---|---|---|---|---|
| DPO (Rafailov et al., 2023) | ✔ | ✘ | ✘ | ✘ | ✔ |
| NPO (Zhang et al., 2024) | ✔ | ✘ | ✘ | ✘ | ✔ |
| WHP (Eldan & Russinovich, 2023) | ✘ | ✔ | ✔ | ✔ | ✘ |
| LLMU (Yao et al., 2024a) | ✔ | ✔ | ✘ | ✘ | ✘ |
| ECO-Prompts (Liu et al., 2024a) | ✘ | ✘ | ✔ | ✘ | ✘ |
| ULMR (Shi et al., 2024a) | ✔ | ✔ | ✔ | ✘ | ✘ |
| FLAT (Wang et al., 2025b) | ✘ | ✘ | ✘ | ✘ | ✔ |
| TPO (Zhou et al., 2025) | ✘ | ✘ | ✘ | ✔ | ✘ |
| TSFD (Kumar, 2025) | ✔ | ✘ | ✔ | ✘ | ✘ |
| DTO(Ours) | ✔ | ✘ | ✘ | ✘ | ✔ |

Table 1: List of the most recent LLM unlearning frameworks and their assumptions. The checkmark (✔) and cross (✘) denote that the framework requires the resource or not.

While several LLM unlearning algorithms have been proposed to unlearn fine-tuning data from a fine-tuned model, their underlying dependence on external resources limits their practicality in real-world scenarios. Table 1 summarizes the external resources that existing LLM unlearning frameworks depends on, and their overall practicalities. A red check mark indicates that the use of the corresponding resource may be impractical. Some works rely on retain set, consisting of the fine-tuning data excluding the forget set, to maintain model utility (Yao et al., 2024a; Wang et al., 2025a). However, assuming access to the retain set may be unrealistic, as data regulations such as GDPR impose strict limitations on storing and reusing raw data (Basaran et al., 2025). Other works assume availability of auxiliary models. These include (1) prompt classifiers (Liu et al., 2024a; Deng et al., 2025), which detect inputs related to the forget data during inference, (2) additional LLMs (Eldan & Russinovich, 2023; Kumar, 2025) obtained by further fine-tuning on the forget set. The auxiliary models might not be available due to the high computational and storage costs of maintaining additional LLMs as well as potential privacy risk that the knowledge of forget set being embedded and persisting in the auxiliary LLMs. Finally, some works leverage existing LLM services. Eldan & Russinovich (2023); Shi et al. (2024a); Zhou et al. (2025) utilized ChatGPT-4 to generate custom dataset from the raw forget set. However, external AI services are generally untrusted as they may collect input data for future training (OpenAI, 2025). Exposing sensitive forget set directly to such services could cause additional privacy risk (Wu et al., 2024). A few works do not require external sources, however they suffer from poor unlearning efficacy or model utility.

**Contributions.** To address these issues, we propose Direct Token Optimization (DTO) for unlearning fine-tuned LLMs, which does not rely on any external resource, such as auxiliary models, retain dataset, or external AI services. Given a token sequence from forget set, DTO identifies the target tokens that are most critical for representing the knowledge of the sequence and utilize them for unlearning. Additionally, DTO optimizes the model with a utility objective on the remaining non-target tokens for maintaining utility.

Different from existing works that rely on human annotator (Yang et al., 2025) or external LLMs such as ChatGPT (Zhou et al., 2025) to select tokens for unlearning, we propose Delta-score, an assistance-free token selection strategy for LLM unlearning inspired by a study of sequence memorization (Stoehr et al., 2024). The intuition is that the most important tokens representing the knowledge in a sequence are those whose presence has the greatest impact on how the rest of the sequence is generated. Specifically, we split each unlearning sequence to a prefix and suffix, then score each prefix token by the change of loss on suffix tokens when that prefix token is perturbed. Then the prefix tokens with highest scores - those with the greatest influence and thus encoding the core knowledge - are chosen as target tokens, on which DTO conducts gradient ascent for unlearning. For the remaining non-target tokens, DTO minimizes the KL-divergence between the logits of the updated and original model to maintain model utility. This ensures that the model forgets the targeted knowledge while preserving its general language ability.

To demonstrate the efficacy of DTO, we conduct experiments with various LLMs on TOFU (Maini et al., 2024) and MUSE (Shi et al., 2024b) benchmarks. We compare DTO to several baselines, including the most recent method FLAT (Wang et al., 2025b), under the same assumptions of relying on only the original model and forget set. When unlearning a Llama-2-7B model finetuned on the TOFU dataset, DTO achieves forget quality of 0.918, a significant improvement over FLAT (Wang et al., 2025b) (0.054). We summarize our contributions as follows.

1. We propose Direct Token Optimization (DTO), a self-contained approach for LLM unlearning that does not require any auxiliary model, retain dataset or external AI services. DTO selects target tokens for unlearning and remaining non-target tokens for utility preservation, and optimizes them accordingly.

2. Inspired by previous studies on memorization, DTO proposes the delta-score for selecting target tokens and non-target tokens, and conducts gradient ascent on target tokens to achieve unlearning while regularizing on non-target tokens to maintain model utility.

3. We conduct experiments and compare the results with state-of-the-art LLM unlearning methods under the same assumptions. The results show that DTO improves the forget quality substantially with only minimal utility degradation.

## 2 RELATED WORK

Machine unlearning was first introduced by Cao & Yang (2015) and has been extensively studied for classification models (Kurmanji et al., 2023; Tarun et al., 2023; Cha et al., 2024; Huang et al., 2024b). This field encompasses two primary approaches. **Exact unlearning** Bourtoule et al. (2021) completely removes the influence of target samples by data partitioning and retraining but often at significant computational cost. **Approximate unlearning** methods iteratively update model parameters so that the unlearned model's behavior approaches that of a retrained model. Notable approaches include maximizing KL-divergence over target sample logits (Kurmanji et al., 2023; Huang et al., 2024b), injecting calibrated noise (Tarun et al., 2023), optimizing the embedding space (Lee et al., 2025), and leveraging adversarial examples (Ebrahimpour-Boroojeny et al., 2025). Certified unlearning refers to approximate unlearning methods that come with certifiable unlearning guarantees (Guo et al., 2019; Koloskova et al., 2025).

Unlearning LLMs is more challenging compared to unlearning classification models. The label space of classification model is usually small and fixed, enabling unlearning through the disassociation of labels of forget set (Kurmanji et al., 2023). The unlearning efficacy and model utility can be directly evaluated by accuracy on forget set and test data (Lee et al., 2025). In contrast, LLMs generate sequences from a vast and unbounded text space, which fundamentally complicates the unlearning process. Unlike classification tasks, unlearning in LLMs cannot be achieved by simple

word suppression (Cooper et al., 2024). Instead, effective unlearning method should minimize the generation of certain sequences or facts (Eldan & Russinovich, 2023; Yao et al., 2024a).

LLM unlearning can be broadly categorized into two types: unlearning pre-trained models and unlearning fine-tuned models. While several methods for unlearning pre-trained models and related benchmarks have been proposed (Li et al., 2024; Jin et al., 2024; Liu et al., 2024b), lack of access to the original pre-training datasets (Yao et al., 2024a) prevents accurate ground truth evaluation and makes scaling to real-world scenarios challenging (Zhou et al., 2024). Unlearning fine-tuned LLMs defines a forget set which was a part of the fine-tuning dataset, and most works are evaluated on benchmarks such as TOFU (Maini et al., 2024) and MUSE (Shi et al., 2024b).

Most works unlearn fine-tuned LLMs by approximate unlearning,such as preference optimization (Zhang et al., 2024; Fan et al., 2024), second-order update (Jia et al., 2024; Gu et al., 2024) and instruction fine-tuning (Shi et al., 2024a). As shown in Table 1, most of them utilize retain set or auxiliary models for better unlearn efficacy and model utility (Yuan et al., 2024; Wang et al., 2025a; Krishnan et al., 2025). However, the availability of retain set or a surrogate dataset with the same distribution Basaran et al. (2025) can not be guaranteed due to privacy regulations, or practical resource limitations (Chundawat et al., 2023). Some works use auxiliary LLM models, obtained by fine-tuning a pre-trained model with the forget set (Wang et al., 2025a; Ji et al., 2024) or further finetuning the finetuned model (Eldan & Russinovich, 2023) with the forget set. However, the knowledge of forget data still remains and the continue fine-tuning introduces extra training cost besides unlearning. Optimization-free methods rely on prompt classifiers to detect forget set inputs and subsequently activate LoRA adapters (Gao et al., 2024; Deng et al., 2025) or corrupt the embeddings (Liu et al., 2024a) to prevent the model from answering the target knowledge. These methods often suffer from limited forget quality, dependency on detection accuracy, and the impracticality of adding prompt classifiers due to training, scalability, and deployment challenges. A more realistic setting is conducting unlearning only with the original model and the forget set. This includes the direct preference optimization (DPO) (Rafailov et al., 2023) and negative preference optimization (NPO) (Zhang et al., 2024). The most recent is FLAT (Wang et al., 2025b), which uses $f$-divergence to steer parameters towards generating refusal responses. While the framework well preserves the model utility, their forget quality shows a clear limitation.

## 3  DIRECT TOKEN OPTIMIZATION

**Problem Definition.** Given a forget set $\mathcal{D}_F$, retain dataset $\mathcal{D}_R$, and an original LLM $\theta_o$ fine-tuned with $\mathcal{D}_F \cup \mathcal{D}_R$ from a pre-trained LLM $\theta_p$, LLM unlearning aims to produce an unlearned model $\theta_u$ that approximates a hypothetical model $\theta_{rt}$ that was fine-tuned only with $D_R$. We assume the unlearner has access only to the forget set $\mathcal{D}_U$ and the original model $\theta_o$, without access to other auxiliary models or retain dataset $D_R$. This setting follows DPO (Rafailov et al., 2023), NPO (Zhang et al., 2024) and FLAT Wang et al. (2025b).

**Intuition.** As shown in Figure 1, the semantic differences between responses from the fine-tuned and retrained model arise primarily from the words highlighted in green and red. This suggests that a small set of tokens are crucial for conveying dataset-specific knowledge. Motivated by this, we aim to unlearn by suppressing the model's ability to generate those crucial tokens while preserving the overall sentence structure that is useful for other queries. We call the crucial tokens as target tokens and the rest as non-target tokens. We propose the delta score to identify target tokens in forget set.

**Delta Score: Identifying Target Tokens.** Prior works (Yang et al., 2025) find that performing unlearning on unique identifier words, such as names and locations, are effective for unlearning samples or entities. However, focusing only on these identifiers is often insufficient, as models may also memorize surrounding context or other tokens that encode the same knowledge. Instead of relying on linguistic rules to pick these identifiers, we adopt a more general token-level perspective, naturally aligning with how LLMs process and memorize unique tokens during fine-tuning (Huang et al., 2024a). This perspective allows our method to target key tokens that contribute to the memorized knowledge, ensuring more effective unlearning. Intuitively, tokens with low per-token-loss indicate stronger memorization and are natural candidates for target tokens as they contribute more to memorizing the sample. However, linguistically important yet semantically uninformative tokens also have low per-token loss, due to the frequent exposure during the pre-training stage (Duan et al.,

2024). Unlearning these causes a detrimental effect on the linguistic fluency of the model. Thus, it is challenging to distinguish target tokens for unlearning from the linguistically important tokens. Most recent works rely on ChatGPT (Zhou et al., 2025) and human annotators (Yang et al., 2025) to identify these tokens, but both approaches are impractical as external AI assistance can be unreliable and untrusted, and human annotation is costly and inconsistent, and both can severely affect the unlearning performance.

Instead, we identify target tokens as the tokens in the prefix that are critical for eliciting the model's memorized output. This is motivated by a study that analyzed memorization in LLMs through perturbation (Stoehr et al., 2024). The study fine-tuned a GPT-Neo model with the PILE dataset (Gao et al., 2020), where each sample is a 100-token paragraph. To quantify memorization, the authors provided the model with a prefix consisting of the first 50 tokens and generated the remainder with greedy decoding; they then measured (1) average NLL over generated sequences and (2) number of generated tokens exactly matching the ground-truth suffix (last 50 tokens of the paragraph). By perturbing one token from the prefix at a time, they identified which token perturbation produced the largest response difference and the biggest NLL spike. Their analysis highlighted that perturbing specific token in the prefix can introduce a significant change in the output, indicating that certain tokens in a prefix serve as a trigger for generating the memorized suffix.

We extend these findings for unlearning purposes. Our insight is that not all tokens in the forget set contribute equally to the model's memorized knowledge; therefore, suppressing this memorized knowledge from the forget set can be most effectively achieved by suppressing the prefix triggers identified by the perturbation analysis. Once these target tokens are identified, taking the gradient ascent on them reduces likelihood of generating themselves, naturally leads to reducing the generation of the memorized knowledge.

While Stoehr et al. (2024) measures NLL over the generated response conditioned on the prefix and partially generated response, our delta-score computes NLL for each suffix token conditioned on the prefix and the original suffix. This gives a stronger signal. When NLL is obtained through conditioning on the *generated* sequence, it naturally decreases toward the end, regardless of whether the response matches the original suffix or not. In contrast, delta-score amplifies the NLL loss by forcing the model to condition on both perturbed prefix and the original suffix. This accurately captures the models' disagreement over the original suffix.

Let $\mathcal{D}_F = \{s^i\}_{i=1}^N$ be a forget set with $N$ samples. Let $s^i = \{x_1^i, \cdots x_t^i \cdots x_{T_i}^i\} \in \mathcal{V}^{T_i}$ be a sequence of tokens $x_t^i$ from the vocabulary $\mathcal{V}$ with the length $T_i$. Let a pivot $1 \leq q_i < T_i$ divide $s^i$ into a prefix $\{x_1^i \cdots x_{q_i}^i\}$ and a suffix $\{x_{q_i+1}^i \cdots x_{T_i}^i\}$. Let a subsequence with size $t-1$ with perturbation on its $r$-th token as $\tilde{x}_{<t}^i := (x_1^i, \ldots, x_{r-1}^i, \tilde{x}_r^i, x_{r+1}^i, \ldots, x_{t-1}^i)$. We define the delta score $\Delta_r^i$ at position $r \leq q_i$ of sequence $s^i$ as follows, and the perturbed token is randomly selected from special tokens ('UNK', '#', etc.).

$$\Delta_r^i = \sum_{t=q_i+1}^{T_i} \left( \log p_\theta \left( x_t^i \mid x_{<t}^i \right) \right) - \sum_{t=q_i+1}^{T_i} \left( \log p_\theta \left( x_t^i \mid \tilde{x}_{<t}^i \right) \right) \tag{1}$$

Equation 1 defines the delta-score as the difference between the average NLL over all suffix tokens when $r$-th token in prefix is present and when it is perturbed. For each unlearning sequence, we select top-$k\%$ highest scoring tokens as target tokens $\mathcal{T}_{k\%}^i = \{x_r^i \mid r \in \text{Top-}k\% \left( \Delta_r^i \right)\}$, and set the rest as non-target tokens $\mathcal{N}^i = s^i \setminus T_{k\%}^i$.

**Unlearning Using Target Tokens.** Target tokens are used for unlearning knowledge from each sequence. We conduct gradient ascent using the loss of predicting the target tokens.

$$\theta_u \leftarrow \theta_u + \eta \nabla_{\theta_u} \sum_{x_t \in \mathcal{T}_{k\%}^i} \log p_\theta \left( x_t \mid x_{<t} \right), \tag{2}$$

where $\eta$ is a step size. Non-target tokens are used for maintaining linguistic fluency and general model utility. For each sequence, we minimize the KL-divergence between the logits of these tokens from the original model $\theta_o$ and the corresponding logits from $\theta_u$.

$$\theta \leftarrow \theta - \nabla_\theta \sum_{x_t \in \mathcal{N}^i} \text{KL}\left( f_{\theta_0} \left( x_{<t} \right) \mid f_{\theta_u} \left( x_{<t} \right) \right) \tag{3}$$

where $f_\theta$ is the model output logit after softmax. While the default DTO is designed to update the model using the non-target tokens, we perform an ablation study in the experiments to compare it with a version of DTO without the KL update. To avoid potential gradient conflicts, each unlearning step (2) and minimizing KL-divergence step (3) are performed in an alternating manner. We orthogonalize one gradient with respect to the other, which further reduces gradient conflicts and improve both unlearn efficacy and model's utility (Kodge et al., 2024) Refer to Appendix C for more details.

## 4 EXPERIMENTS

### 4.1 EXPERIMENTAL SETUP

**Datasets & Evaluation Metrics.** We evaluate our method on two LLM unlearning benchmarks. The TOFU (Maini et al., 2024) dataset has 4,000 question and answer pairs of fictitious authors for finetuning any LLMs. It provides multiple tasks of unlearning 1%, 5% and 10% of the training dataset. The dataset provides following evaluation metrics: **Model utility**, obtained from the aggregated score of Rouge-L, normalized probability over answers, and truth ratio (probability of correct answer over the incorrect answer) on question and answer pairs of remaining, real authors and real world dataset; and **Forget quality**, measured using the Kolmogorov-Smirnov test on the truth ratios using both unlearned and retrained model. A successfully unlearned model should exceed 0.05 Mekala et al. (2024). We also show the Forget-Rouge as a proxy to evaluate the memorization of the model. We use finetuned version provided by authors for Llama 3.2-1B and Llama 2-7B model (Touvron et al., 2023). For MUSE benchmark (Shi et al., 2024b), we use the MUSE-book, which consists of Chapter 2 of the Harry Potter series. Muse benchmark offers following evaluation metrics: **Verbatim Memorization (VerbMem)**, obtained via Rouge-L F1 scores, shows the exact sequence memorization from the model; **Knowledge Memorization (KnowMem)**, obtained via Rouge-L scores on question and answer evaluation dataset, evaluates how the model retains factual knowledge of the unlearning contents; and **Privacy Leakage (PrivLeak)**, obtained via membership inference attack (Carlini et al., 2022a), evaluates if the model's response after unlearning still reveals that the forget set was part of their fine-tuning data. It is a normalized AUC difference of the membership inference attack on the unlearned model and the retrained model. The negative value means the model is under-unlearned, and positive means over-unlearned. The ideal model should have the value close to zero.

**Baselines.** We compare our framework with baselines that have the same assumption (access to forget set and original model only). NPO (Zhang et al., 2024) unlearns by conducting preference optimization to reject answering questions in the forget set. DPO (Rafailov et al., 2023) unlearns by up-weighting generation of a rejection template for the forget set. FLAT (Wang et al., 2025b) is the most recent and state-of-the-art framework that steers the model to generate rejection template over the original answer by maximizing $f$-divergence[1]. As proposed from the paper, we use Kullback–Leibler (KL), Total Variation (TV), Jensen–Shannon (JS) and Pearson (P) divergences. We search hyper-parameters to find the best model utility and forget quality tradeoff for each baseline.

While not directly comparable, we also include LLMU (Yao et al., 2024b) which requires retain dataset. It conducts gradient ascent on all responses and minimizes the KL-divergence between the original model and the unlearned model over the retain set for model utility. In addition, we compare our token selection strategy and the unlearning result with Token Preference Optimization (TPO) Zhou et al. (2025) which uses ChatGPT to identify target words to unlearn.

### 4.2 EXPERIMENTAL RESULTS

**Unlearn Efficacy and Model Utility.** Table 2 shows the result of unlearning 1% of the TOFU dataset with DTO and baselines. We used top $k = 20\%$ for selecting target tokens and suffix ratio of 0.25, or last 25% of each sequence as a suffix. DTO without KL achieves the highest and almost perfect forget quality of 0.9188 compared to an ideal retrained model, while most baselines remain below 0.055 and 0.16. This is a significant margin, indicating DTO without KL is extremely

---

[1]The official implementation was inaccessible, hence we re-implemented this baseline, and confirmed that the result is comparable. Refer to Appendix D for details.

| | Llama2-7B | | | Llama3.2-1B | | |
| --- | --- | --- | --- | --- | --- | --- |
| | Forget Quality (↑) | Model Utility (↑) | Forget Rouge-L (↓) | Forget Quality (↑) | Model Utility (↑) | Forget Rouge-L (↓) |
| Original LLM | 2.183e-06 | 0.6346 | 0.8849 | 3.383e-06 | 0.6218 | 0.8168 |
| Retrained LLM | 1.0000 | 0.6267 | 0.4080 | 1.0000 | 0.6168 | 0.4045 |
| LLMU | 0.0541 | 0.6225 | 0.4472 | 0.0301 | 0.5876 | 0.4671 |
| DPO | 0.0541 | 0.6219 | 0.5724 | 0.0541 | 0.5191 | 0.4752 |
| NPO | 0.0068 | 0.6242 | 0.4523 | 0.2650 | 0.5608 | 0.2447 |
| FLAT (TV) | 0.0541 | 0.6199 | 0.4366 | 0.1649 | 0.5687 | 0.2543 |
| FLAT (KL) | 0.0301 | 0.6393 | 0.4971 | 0.1430 | 0.5639 | 0.2688 |
| FLAT (JS) | 0.0970 | 0.6214 | 0.4252 | 0.0286 | 0.5548 | 0.3793 |
| FLAT (P) | 0.0541 | 0.6239 | 0.4523 | 0.1649 | 0.5578 | 0.2567 |
| **DTO w/o KL (Ours)** | **0.9188** | 0.5948 | **0.3725** | **0.9188** | 0.5375 | 0.2893 |
| **DTO (Ours)** | **0.7659** | 0.6002 | 0.3978 | **0.7659** | 0.5529 | **0.2382** |

Table 2: Experimental results on TOFU 1% dataset.

| | Llama 2-7B | | | Llama 3.2-1B | | |
| --- | --- | --- | --- | --- | --- | --- |
| | Forget Quality (↑) | Model Utility (↑) | Forget Rouge-L (↓) | Forget Quality (↑) | Model Utility (↑) | Forget Rouge-L (↓) |
| Original LLM | 4.513e-09 | 0.6319 | 0.8938 | 4.525e-08 | 0.6218 | 0.8250 |
| Retrained LLM | 1.0000 | 0.6263 | 0.3982 | 1.0000 | 0.6098 | 0.3857 |
| LLMU | 1.143e-05 | 0.3193 | 0.2310 | 0.0001 | 0.5928 | 0.6975 |
| DPO | 5.617e-06 | 0.4962 | 0.4857 | 0.0005 | 0.4966 | 0.4934 |
| NPO | 4.744e-06 | 0.5906 | 0.3977 | 0.0007 | 0.5536 | 0.4008 |
| FLAT (TV) | 0.0021 | 0.1253 | 0.0534 | 0.0124 | 0.2071 | 0.0988 |
| FLAT (KL) | 2.353e-05 | 0.2402 | 0.2832 | 0.0878 | 0.3266 | 0.1398 |
| FLAT (JS) | 0.0001 | 0.3091 | 0.1716 | 0.0001 | 0.4514 | 0.2118 |
| FLAT (P) | 1.873e-05 | 0.1971 | **0.0825** | 0.0030 | 0.2268 | **0.0902** |
| **DTO w/o KL (Ours)** | 0.0021 | 0.2871 | 0.1743 | 0.0001 | 0.3187 | 0.2597 |
| **DTO (Ours)** | **0.0876** | 0.4442 | 0.5415 | **0.3281** | 0.4218 | 0.3168 |

Table 3: Experimental results on TOFU 5% dataset

effective in unlearning the target knowledge. DTO preserved model utility better by minimizing KL-divergence of logits on non-target tokens between unlearned model and original model. However, the forget quality is slightly lower. Moreover, DTO without KL is exhibiting the lowest Rouge-L score from the forget set (0.3725). This confirms that the actual response from the unlearned model is significantly less similar to the response of the forget set. While both DTO and DTO without KL are showing model utility drop, the degradation from the original model is small, demonstrating a reasonable tradeoff given its strong forget quality. FLAT exhibits better model utility, but their forget quality is low, indicating inherent knowledge about forget set still persists in the model. Both LLMU, DPO and NPO shows good model utility,

| | VerbMem on $D_u$ (↓) | KnowMem on $D_u$ (↓) | KnowMem on $D_r$ (↑) | PrivLeak (↓) |
| --- | --- | --- | --- | --- |
| Original Model | 99.70 | 45.87 | 68.40 | -58.19 |
| LLMU | 99.70 | 44.60 | 67.69 | -57.37 |
| DPO | 46.95 | 41.28 | 65.24 | -57.24 |
| NPO | 68.85 | 30.65 | 48.96 | -53.90 |
| FLAT (TV) | 99.13 | 40.54 | 59.63 | -57.51 |
| FLAT (KL) | 99.70 | 44.07 | 63.41 | -57.55 |
| FLAT (JS) | **15.84** | 25.59 | 49.85 | -47.27 |
| FLAT (P) | 98.22 | 43.00 | 60.89 | -57.59 |
| **DTO w/o KL (Ours)** | 19.30 | **22.84** | 57.11 | **-47.14** |
| **DTO (Ours)** | 88.42 | 46.21 | 66.40 | -57.22 |

Table 4: Evaluation results on MUSE-Book dataset. Unlearn efficacy is assessed by VerbMem on $D_f$ (↓), KnowMem on $D_f$ (↓), and PrivLeak (↓). Model utility is assessed by KnowMem on $D_r$ (↑).

Table 3 shows the result of unlearning 5% of the TOFU dataset with DTO and baselines. LLMU, DPO and NPO have relatively high model utility and low forget quality. FLAT is showing both

lower model utility and forget quality, implying over-unlearning. When the model loses linguistic capability due to harsh optimizations, it provides random answers for the QA dataset and shows different output distribution as the retrained model, resulting low forget quality. Moreover, DTO shows better forget quality and better model utility than DTO without KL. By minimizing KL-divergence between original model over the logits of non-target tokens, the unlearned model was able to prevent losing linguistic capability, which leads to better forget quality. Compared to the baselines, DTO achieved best forget quality and relatively high model utility. For results of unlearning 10% TOFU data, please refer to Appendix E. Overall, DTO achieves the best forget quality over the TOFU dataset with comparable model utility.

Table 4 shows the reuslt of unlearning MUSE-Books with DTO and baselines. LLMU, DPO and NPO failed to remove the verbatim memorization from forget set. Except for FLAT (JS), every other FLAT variants failed to remove the memorization. DTO w/o KL shows relatively small verbatim memorization. This was achievable because DTO directly suppresses tokens that trigger verbatim memorization. KnowMem on forget set evaluates more intrinsic knowledge memorization of forget set with QA datasets. DTO w/o KL achieved the lowest, showing that it also is capable of removing intrinsic knowledge. Similarly, KnowMem on retain data shows that DTO is able to keep the rest of the knowledge relatively intact. Lastly, PrivLeak shows the normalized membership inference risk compared to the retrained model. The negative sign means that the risk persists, and a score closer to zero means less risk. DTO w/o KL shows the closest score to zero among all baselines.

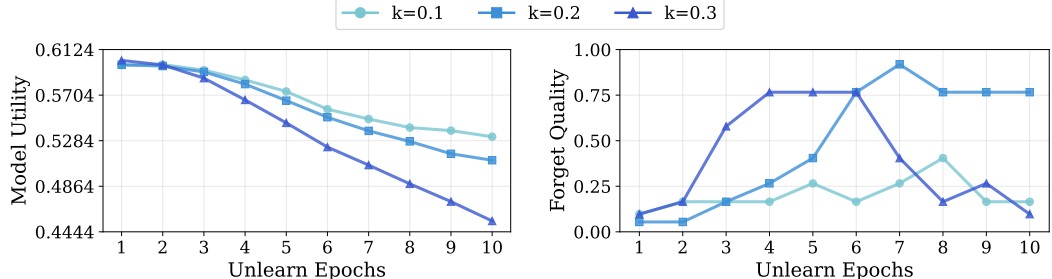

Figure 2: Model Utility and Forget quality with respect to various $k$. Suffix ratio is fixed to 0.25

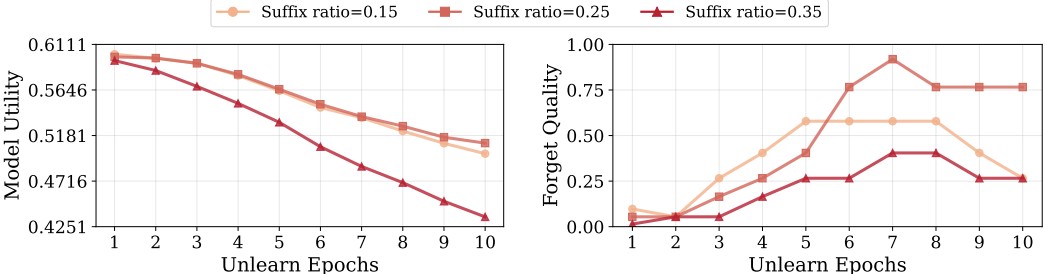

Figure 3: Model Utility and Forget quality with respect to various suffix ratios. $k$ is fixed to 0.2

**Parameter Studies of Target Token Ratio and Suffix Ratio.** Figure 2 shows the progression of unlearning 1% of TOFU dataset over various target token ratio $k$ with fixed suffix ratio. When $k = 0.1$, top-10% tokens with the highest delta-scores are selected as target tokens. Smaller size of the target tokens preserves model utility well, however, forget quality hardly increases, meaning that top-10% were insufficient to unlearn. On the other hand, when $k = 0.3$, the utility of the model drops rapidly and the quality of the forget increases quickly. However, after 6th epoch, forget quality starts to drop due to over-unlearning.

Figure 3 shows the progression of unlearning 1% of TOFU dataset over various suffix ratios with fixed $k$. Delta-score for each prefix token is obtained from average NLL loss of suffix tokens. This makes the choice of suffix ratio critical. When suffix ratio is 0.15

(last 15% of the tokens), model utility is largely preserved yet forget quality is less optimal. This is because a small suffix may contain only a limited portion of the target knowledge, providing insufficient signal for delta score to identify the most critical prefix tokens. Thus the selected target tokens from the prefix might not fully suppress the generation of target knowledge. Conversely, when suffix ratio is too large, irrelevant tokens start to influence the delta-score, introducing noise that leads to lower model utility and forget quality.

```
<|begin_of_text|>, [/, INST, ], What, gender, is,
 author, Basil, Mah, f, ouz, Al, -K, u, wait, i,
?, [/, INST, ], Author, Basil, Mah, f, ouz, Al,
-K, u, wait, i, is, male, .[, /, INST, ],
<|eot_id|>
```

(a) tokens selected by Delta-Score (Ours)

```
<|begin_of_text|>, [/, INST, ], What, gender, is,
 author, Basil, Mah, f, ouz, Al, -K, u, wait, i,
?, [/, INST, ], Author, Basil, Mah, f, ouz, Al,
-K, u, wait, i, is, male, .[, /, INST, ],
<|eot_id|>
```

(b) tokens selected by ChatGPT

```
<|begin_of_text|>, [/, INST, ], What, specific,
 genre, is, Nikol, ai, Ab, il, ov, known, for,
?, [/, INST, ], N, ik, ol, ai, Ab, il, ov, is,
 most, celebrated, for, his, compelling,
 writing, in, the, African, American, genre,
,, bringing, fresh, perspectives, through,
 his, unique, cultural, lens, .[, /, INST, ],
<|eot_id|>
```

(c) tokens selected from by Delta-Score (Ours)

```
<|begin_of_text|>, [/, INST, ], What, specific,
 genre, is, Nikol, ai, Ab, il, ov, known, for,
?, [/, INST, ], N, ik, ol, ai, Ab, il, ov, is,
 most, celebrated, for, his, compelling,
 writing, in, the, African, American, genre,
,, bringing, fresh, perspectives, through,
 his, unique, cultural, lens, .[, /, INST, ],
<|eot_id|>
```

(d) tokens selected from by ChatGPT

Figure 4: Selected tokens from sample 2 (first row) and 26 (second row) by the Delta-score and TPO

**Token Selection Strategy.** We compare our delta-score with the token selection strategy of Token Preference Optimization (TPO) Zhou et al. (2025), which uses ChatGPT to identify target *words*, and use preference optimization to unlearn. We follow the instruction provided in the paper and prompt ChatGPT with the TOFU dataset. Figure 4 shows the tokens selected by Delta-score and ChatGPT. Figure 4a and 4b shows that the delta-score primarily identified the last name of the person (Mahfouz Al-Kuwaiti) as targets similar to ChatGPT's selection, indicating that crucial tokens can be selected without external AI services. Figure 4c and 4d show that delta-score selected the name (Nikolai Abilov) and "celerbrated" while ChatGPT picked the specific genre. The delta-score could not directly select the genre since it was part of the suffix. Instead, it selected the tokens that are directly relatable to the genre. Table 5 shows the result of unlearning 1% of TOFU dataset from Llama 3.2-3B using DTO and TPO (Zhou et al., 2025)[2]. Although DTO exhibits lower forget quality (which could be due to less than perfect target token selection and gradient ascent based unlearning), our token selection method avoids privacy risks associated with untrusted external AI services. In addition, reducing the suffix length further improves DTO's selection correctness, as indicated by Figure 3. Finally, our selection strategy is general and can use preference optimization instead of gradient ascent to enhance the unlearning.

| | Forget Quality | Model Utility |
|---|---|---|
| TPO | 0.55 | 0.61 |
| DTO | 0.26 | 0.42 |

Table 5: Model Utility % Forget Quality of unlearning 1% of TOFU dataset from Llama 3.2-3b

## 5  CONCLUSION

In this paper, we proposed Direct Token Optimization (DTO), a self-contained unlearning framework that unlearns an LLM without external resources, such as external reference models and retain datasets. Given a sequence to unlearn, DTO identifies target tokens that trigger the memorized knowledge using the proposed delta-score. During unlearning, target tokens are used for unlearning optimization while non-target tokens are used for maintaining model utility. Experimental results show that the DTO achieves substantially better forget quality than the state-of-the art methods while retaining reasonable model utility. In future work, we aim to incorporate preference optimization to improve the model utility forget quality trade-off, and further improve delta-score to more effectively select target tokens in each sequence.

---

[2]We compare the result of TPO that is reported from the paper.

## REPRODUCIBILITY STATEMENTS

We use the official dataset provided from TOFU (Maini et al., 2024) and MUSE (Shi et al., 2024b) benchmarks. For evaluations we use the official code provided from Open-Unlearning (Dorna et al., 2025), a public github repository that provide comprehensive evaluation framework for these datasets. For unlearning TOFU data, we use the Llama-2-7B, Llama 3.2-1B, and Llama 3.2-3B models fine-tuned on the TOFU dataset, available in Dorna et al. (2025) and Maini et al. (2024). For MUSE, we use Llama-2-7B model officially fine-tuned on Chapter 2 of the Harry Potter series, available at Shi et al. (2024b). We offer our code and result files in this link to the anonymous git repository. We provide detailed hyperparameter settings in Appendix B.

## ETHICS STATEMENTS

Our proposed approach contributes to society and to human well-being by illustrating limitations and possible privacy risk of existing LLM unlearning methods, and proposing a novel machine unlearning method, which can protect individuals' data privacy in more rigorous manner. We strictly comply the code of ethics to ensure all data is properly handled, and conducted fair experiments.

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

## APPENDIX

In this appendix session A describes the use of large language models in this research. Section B provides detailed hyperparameter settings. Section C illustrates gradient orthogonalization. Section D discusses the integrity of our baseline implementation. Section E illustrates experimental results on unlearning TOFU 10% dataset.

## A LLM USAGE

An LLM has been partially contributed to this research. LLMs assisted resolving minor technical issues on implementing baselines, our proposed method and experiments testbed. We rarely used LLM for assisting writing. While the LLM provided some writing suggestions, we did not directly copy and paste the LLM generated paragraphs into the paper.

## B HYPERPARMETER DETAILS AND IMPLEMENTATIONS

| forget set | Model | $k$ | Suffix ratio | Batch size | Learning rate |
|---|---|---|---|---|---|
| TOFU 1% | Llama 3.2-1b | 0.2 | 0.25 | 8 | $1e-5$ |
| | Llama 2-7b | 0.2 | 0.25 | 8 | $1e-5$ |
| TOFU 5% | Llama 3.2-1b | 0.2 | 0.2 | 8 | $2e-5$ |
| | Llama 2-7b | 0.15 | 0.15 | 8 | $1e-5$ |
| TOFU 10% | Llama 3.2-1b | 0.1 | 0.1 | 8 | $5e-5$ |
| | Llama 2-7b | 0.1 | 0.15 | 8 | $2e-5$ |
| Muse-Books | Llama 2-7b | 0.2 | 0.25 | 8 | $1.5e-6$ |

Table 6: List of hyperparameters for DTO

Table 6 shows the list of hyperparameters we used for each of the dataset. We used the same set of parameters for DTO and DTO without KL. For more details on implementations, please refer to this link to the anonymous git repository.

## C GRADIENT ORTHOGONALIZATION

Gradient from unlearning loss and forget set and gradient from retain loss and retain set often has conflict. The directions often interfere themselves. Naively conducting each step-wise update leads to imperfect optimization for both objectives. This leads to a catastrophic utility loss. Gradient orthogonalization reduces the gradient conflict by projecting a gradient (Kodge et al., 2024). Given two gradients $g_a$ and $g_b$, orthogonalizing $g_a$ to $g_b$ computs followings:

$$g_a^{orth} = g_b - \frac{\langle g_a, g_b \rangle}{\langle g_a, g_b \rangle} g_b. \qquad (4)$$

Intuitively, this nullifies optimization directions in $g_a$ that are parallel to $g_b$, allowing less impact on the objective of $g_b$. In our method, we orthogonalize gradients from unlearn loss to the gradient of the retain loss, to achieve unlearn objective with less detrimental impact on model utility.

## D BASELINE IMPLEMENTATIONS

DPO, NPO and LLMU have official implementations, however, FLAT (Wang et al., 2025b) is missing the implementation, hence we implemented them based on the paper. To verify integrity of our implementation, we compare our unlearning results with the reported results of FLAT. Table 7 compares the result of unlearning 1% of TOFU dataset on LLama 2-7B. Although model utility is slightly lower, the results show that our implementation is consistent with the reported results.

| | Our implementation | | | Reported in Wang et al. (2025b) | | |
|---|---|---|---|---|---|---|
| | Forget Quality ($\uparrow$) | Model Utility ($\uparrow$) | Forget Rouge-L ($\downarrow$) | Forget Quality ($\uparrow$) | Model Utility ($\uparrow$) | Forget Rouge-L ($\downarrow$) |
| Original LLM | 2.183e-06 | 0.6346 | 0.8849 | 4.488e-06 | 0.6346 | 0.9851 |
| Retrained LLM | 1.0000 | 0.6267 | 0.4080 | 1.0000 | 0.6267 | 0.0.4080 |
| FLAT (TV) | 0.0541 | 0.6199 | 0.4366 | 0.0541 | 0.6373 | 0.4391 |
| FLAT (KL) | 0.0301 | 0.6393 | 0.4971 | 0.0286 | 0.6393 | 0.5199 |
| FLAT (JS) | 0.0970 | 0.6214 | 0.4252 | 0.0541 | 0.6364 | 0.4454 |
| FLAT (P) | 0.0541 | 0.6239 | 0.4523 | 0.0541 | 0.6374 | 0.4392 |

Table 7: Experimental result on TOFU 1% dataset of FLAT implemented by us and the reported results.

| | Llama 2-7B | | | Llama 3.2-1B | | |
|---|---|---|---|---|---|---|
| | Forget Quality ($\uparrow$) | Model Utility ($\uparrow$) | Forget Rouge-L ($\downarrow$) | Forget Quality ($\uparrow$) | Model Utility ($\uparrow$) | Forget Rouge-L ($\downarrow$) |
| Original LLM | 1.735e-08 | 0.6346 | 0.8824 | 3.382e-06 | 0.6218 | 0.8194 |
| Retrained LLM | 1.0000 | 0.6122 | 0.3998 | 1.0000 | 0.5936 | 0.3785 |
| LLMU | 1.092e-06 | 0.2903 | 0.1127 | 4.353e-05 | 0.5795 | 0.6501 |
| DPO | 1.826e-07 | 0.5178 | 0.5745 | 1.119e-07 | 0.4764 | 0.4845 |
| NPO | 1.065e-06 | 0.5326 | 0.3587 | 1.839e-06 | 0.5429 | 0.4293 |
| FLAT (TV) | 4.35e-05 | 0.0866 | 0.0267 | 0.0013 | 0.2299 | 0.1367 |
| FLAT (KL) | 5.418e-05 | 0.0219 | 0.0013 | 0.0013 | 0.2727 | 0.1648 |
| FLAT (JS) | 4.587e-05 | 0.0802 | 0.0365 | 3.277e-05 | 0.3702 | 0.2952 |
| FLAT (P) | 0.0001 | 0.0538 | 0.0148 | 0.0005 | 0.1639 | 0.1089 |
| **DTO w/o (Ours)** | 3.913e-06 | 0.1623 | 0.2118 | 4.353e-05 | 0.1971 | 0.2397 |
| **DTO (Ours)** | **0.0004** | 0.4507 | 0.5580 | **0.0365** | 0.3067 | 0.3477 |

Table 8: Experimental results on TOFU 10% dataset

# E  ADDITIONAL EXPERIMENTS

Table 8 shows the result of unlearning 10% of the TOFU dataset with DTO and baselines. For both models, LLMU, DPO and NPO failed to eliminate unlearn knowledge. FLAT achieved better forget quality, however, they suffer significaint utility loss. The utility loss is more significant from the 7B model than 1B model. We assume that $f$-divergence of FLAT over-generalizes the rejection template when the number of unlearning samples increases. DTO successfully reduced model utility loss, while achieving the best forget quality.

While DTO achieved the best forget quality among all baselines, forget quality failed to exceed 0.05, which serves as a statistical threshold for successful unlearning (Mekala et al., 2024). Due to the size of the dataset, unlearning TOFU 10% is challenging without the retain dataset. We aim to improve this in our future studies.

