# OpenReview forum: "Direct Token Optimization: A Self-contained Approach to Large Language Model Unlearning"
_ICLR.cc/2026/Conference — ICLR 2026 Conference Withdrawn Submission_

### Official Review · Reviewer_Cyf2 · 2025-10-22

**Soundness:** 2
**Presentation:** 3
**Contribution:** 2
**Rating:** 4
**Confidence:** 4

**Summary:**

This paper proposes Direct Token Optimization (DTO), a self-contained method for unlearning knowledge from fine-tuned large language models (LLMs) without using retain datasets, auxiliary models, or external APIs. The core idea is to identify “target tokens”—tokens most responsible for encoding the forget-set knowledge—using a new metric called delta-score, which measures the influence of each prefix token on predicting the suffix. DTO performs gradient ascent on these target tokens to erase memorized knowledge, while preserving general utility by minimizing KL divergence on the remaining non-target tokens. Experiments on TOFU and MUSE benchmarks show that DTO achieves significantly higher forget quality than prior methods such as FLAT, DPO, NPO, and LLMU, while maintaining reasonable utility, all under a more realistic resource-constrained setting.

**Strengths:**

1. The method does not rely on retain data, auxiliary models, or external APIs; it only requires the forget set, making it more practical.
2. The method performs targeted, token-level updates, focusing only on tokens that encode forget-related knowledge. This minimizes collateral damage to unrelated knowledge and helps preserve the model’s overall capabilities.
3. The experimental evaluation is thorough, spanning multiple datasets and metrics, and provides convincing evidence of both the method’s effectiveness and its reliability.

**Weaknesses:**

1. The idea of token-level optimization is not entirely new and appears to be an incremental refinement of TPO [1], rather than a fundamentally different approach.
2. The paper lacks analysis of computational complexity and does not provide experiments on spatial or temporal efficiency (e.g., training time, GPU memory usage), which makes it difficult to assess the method’s scalability.
3. Although the target token selection is improved, treating all remaining tokens uniformly as “utility-preserving language tokens” seems overly simplistic. This assumption ignores that some of these tokens may also contribute to task-relevant knowledge, and thus model utility may not be preserved in a principled way.
4. The method relies heavily on the assumption that forget-related knowledge can be localized through prefix–suffix structures. This may fail when answers are short phrases instead of complete sentences, when the expected ordering of prefix and suffix does not hold, or when linguistic structure is more complex. Moreover, the effectiveness of target token selection is sensitive to hyperparameters, such as the prefix length and the defined delta-score ratio.

[1] Zhou, X., Qiang, Y., Zade, S. Z., Zytko, D., Khanduri, P., & Zhu, D. (2025). Not All Tokens Are Meant to Be Forgotten. arXiv preprint arXiv:2506.03142.

**Questions:**

1. Have you evaluated this approach when applied to other unlearning methods such as NPO [1] or SimNPO [2]? If the method is compatible in principle with forget-only optimization, how does it perform in those settings?
2. What is the time and computational complexity of the proposed method? Specifically:
(1) What is the complexity of identifying target tokens?
(2) What is the additional training overhead of optimizing only on these tokens?
(3) Are there experiments measuring training time, GPU memory usage, or scaling behavior?
If target token selection is a core component, more detailed ablations or efficiency analyses would be valuable.
3. How would this method extend to multimodal LLMs (MLLMs)? When visual information is integrated (e.g., via cross-attention with images), does target token selection need to be adapted? Would the prefix–suffix assumption still hold in such a setting?

[1] Zhang, R., Lin, L., Bai, Y., & Mei, S. (2024). Negative preference optimization: From catastrophic collapse to effective unlearning. arXiv preprint arXiv:2404.05868.
[2] Fan, C., Liu, J., Lin, L., Jia, J., Zhang, R., Mei, S., & Liu, S. (2024). Simplicity prevails: Rethinking negative preference optimization for llm unlearning. arXiv preprint arXiv:2410.07163.

---

> ### Author Response · Authors · 2025-11-21
> **Response to reviewer Cyf2**
>
> We thank the reviewer for the considerate and detailed review. We provide clarification for each item.
>
> **Weakness 1**
> The token-level optimization is widely used for various language modeling tasks, including RLHF reward modeling [1] and preference optimization [2]. This is used in both TPO and our DTO, but the our token selection is fundamentally different. TPO simply out-sourced the token identification process to ChatGPT, depending on potentially untrusted external resources, provides no transparency why certain tokens are selected, and lacks consistency or reproducibility. In contrast, we choose the target and non-target tokens via $\Delta$-score, avoiding relying on external resources, and providing a more fundamental insight on why and which tokens are critical for unlearning.
>
> **Weakness 2 & Question 2**
> DTO involves two processes: token identification and unlearning. The time complexity of token identification scales linearly with the dataset size. Let $N$ the dataset size and $s$ the suffix size ($0<s<1$). The time complexity of token identification is $O(100(1-s)N)$, as it needs $100(1-s)$ forward passes to identify target tokens from a sequence. The time for the unlearning process is identical to baselines, as both use the same model, batch-size and number of iterations. We provide the processing time (in seconds) for token identification for different splits of TOFU dataset. The table shows that time overhead from the token identification is not significant and scales well with the model and unlearning data size.
>
> |Split|1%|5%|10%|
> |-|-|-|-|
> |Llama 3.2-1B|17.21|34.62|164.18|
> |Llama 2-7B|40.12|112.73|566.32|
>
> The following table shows the memory usage (GB) during unlearning Llama 1B model with DTO and a baselines for unlearning different portions of TOFU dataset. It shows that it’s comparable to the baseline and scales well with the unlearning data size.
>
> |Method|1%|5%|10%|
> |-|-|-|-|
> |DTO|24.06|37.86|67.18|
> |FLAT|33.05|58.18|58.18|
>
> For scaling behavior, we provide the running time (minutes) and memory usage (GB) of DTO for unlearning 1% of TOFU dataset on models from 1B to 7B. It suggests that it scales sublinearly with the model size.
>
> |Model size|Runtime (minutes)|Memory (GB)|
> |-|-|-|
> |1B|3.98|24.06|
> |7B|12.33|101.17|
>
> **Weakness 3**
> We do not use a specific criteria to select *“tokens for utility”* because we aim to preserve the general capability of the target model. The original model in our evaluation aims to keep a general language modeling capability across tasks (question answering, multiple choice questions, etc.). If the original model is specialized to a specific task, we could use the task-dependent optimization instead of KL regularization to enhance model utility
>
> **Weakness 4**
> DTO does not rely on strict linguistic prefix-suffix structures. The TOFU dataset includes question and answer pairs of various formats and lengths, ranging from short phrases to multi-sentence responses. We choose the prefix and suffix purely on the ratio of the sequence instead of relying on specific rules. The suffix may include part of the question, part of the answer, or both. Across all such cases, DTO consistently achieves effective unlearning, indicating that it is robust to the linguistic structure of the sequence.
>
> DTO is not sensitive to the choice of target token ratio (k%) and suffix ratio. Table 6 in the Appendix shows that the optimal $k$ is consistently between a narrow range of 0.1 and 0.2, and the optimal suffix ratio between 0.1 to 0.25 across different model sizes and datasets. This small and stable range shows that the search space of the hyperparameters is small and the tuning can be done efficiently. Moreover, the performance degrades smoothly outside the optimal region, meaning an approximate value (e.g., $k$=0.15) already yields strong results without dataset-specific fine-tuning.
>
> **Question 1**
> NPO and simNPO use sequence-level preference optimization. DTO uses token-level optimization, hence cannot be directly plugged into NPO/simNPO without reformulating the optimization at the token level. Instead, DTO is compatible with token-level preference optimization [2] (partially used by TPO). Integrating DTO with token-level preference optimization is a promising direction for future work, but it is an orthogonal extension and does not diminish the core contribution of our method: a principled token-selection mechanism for unlearning.
>
> **Question 3**
> We appreciate the suggestion on applying DTO in multi-model LLMs. While our general idea may apply, it would require dedicated methodological development to account for the cross-modality interactions. We consider this as a promising direction for future work but beyond the scope of the present paper.
>
> ***
>
> [1] Zhou et al., "T-REG: Preference Optimization with Token-Level Reward Regularization,” ACL 2025
>
> [2] Zeng et al., “Token-Level Direct Preference Optimization,” ICML 2024

---

### Official Review · Reviewer_69ut · 2025-10-26

**Soundness:** 2
**Presentation:** 2
**Contribution:** 2
**Rating:** 4
**Confidence:** 5

**Summary:**

This paper introduces Direct Token Optimization (DTO), a self-contained method for machine unlearning in large language models (LLMs). Unlike prior work that depends on auxiliary models, retain datasets, or external AI services (e.g., ChatGPT) to remove the influence of a “forget set,” DTO performs unlearning directly on the original model using only the forget data. The core idea is to identify target tokens—those most responsible for encoding the memorized knowledge—using a novel metric called the delta-score, which measures the impact of perturbing each token on the loss of subsequent tokens. DTO performs gradient ascent on these target tokens to “forget” them while preserving general linguistic ability by minimizing KL-divergence between logits of non-target tokens in the unlearned and original models.

Experiments on the TOFU and MUSE benchmarks demonstrate that DTO achieves up to 16.8× improvement in forget quality compared to recent baselines (e.g., FLAT, DPO, NPO), while maintaining comparable utility. The method shows strong empirical performance without relying on any external resources, making it practical and privacy-preserving.

**Strengths:**

1. Novel and Practical Approach. DTO is the fully self-contained LLM unlearning method that does not require retain sets, auxiliary models, or external AI services. The token-level optimization framework and the introduction of the delta-score for selecting influential tokens are conceptually intuitive.
2. Balanced Optimization Design. The two-stage optimization—gradient ascent on target tokens and KL minimization on non-target tokens—provides a clear mechanism to balance unlearning and utility.
3. Comprehensive Experiments and Clear Gains. Evaluations across TOFU and MUSE benchmarks are extensive and demonstrate significant improvements in forget quality while maintaining reasonable model utility. Ablation and parameter studies (e.g., Figure 2–3) effectively illustrate the effects of target ratio k and suffix ratio on the unlearning trade-off.

**Weaknesses:**

1. Limited Theoretical Grounding. The paper provides little theoretical analysis of delta-score properties or convergence guarantees of the optimization. The method remains primarily empirical and intuitive.
2. Scalability and Generalization. Experiments are limited to small and mid-sized models (LLaMA 2–7B, LLaMA 3.2–3B). It is unclear whether DTO scales efficiently to larger production-scale LLMs (e.g., 70B+).
3. Heuristic Hyperparameter Choices. The selection of target token ratio (k%) and suffix ratio is based on manual tuning. The method may require domain-specific adjustments, reducing general applicability.
4. Limited Qualitative Analysis. Although quantitative metrics are strong, more qualitative examples of “before-and-after unlearning” would help demonstrate DTO’s linguistic preservation and knowledge removal in practice.
5. Some relevant prior works should be included and discussed [1-3].


[1] Liu, Sijia, Yuanshun Yao, Jinghan Jia, Stephen Casper, Nathalie Baracaldo, Peter Hase, Yuguang Yao et al. "Rethinking machine unlearning for large language models." Nature Machine Intelligence (2025): 1-14.

[2] Tang, Haoyu, Ye Liu, Xi Zhao, Xukai Liu, Yanghai Zhang, Kai Zhang, Xiaofang Zhou, and Enhong Chen. "Learn while unlearn: An iterative unlearning framework for generative language models." arXiv preprint arXiv:2407.20271 (2024).

[3] Patel, Gaurav, and Qiang Qiu. "Learning to unlearn while retaining: Combating gradient conflicts in machine unlearning." In Proceedings of the IEEE/CVF International Conference on Computer Vision, pp. 4211-4221. 2025.

**Questions:**

1. Is there a way to adaptively determine the k% target ratio or suffix ratio during training rather than manual tuning?
2. See the weaknesses.

---

> ### Author Response · Authors · 2025-11-21
> **Response to reviewer 69ut**
>
> We thank the reviewer for a considerate and detailed review. We provide clarification for each item.
>
> **Weakness 1.**
> We agree that a deeper theoretical analysis would further strengthen our framework. However, deriving closed form convergence guarantee or an analytical characterization of the delta-score is infeasible for highly nonlinear and nonconvex models such as LLMs. This limitation is not particular to our work: most existing LLM unlearning approaches lack provable convergence results due to the complexity of the optimization landscape. Our delta-score is constructed to approximate the contribution of a token. While formal guarantees of LLM unlearning remain an open challenge for the field, our empirical results provide strong evidence that 1) the delta-score based token selection is effective in unlearning while maintaining model utility, 2) the method converges stably across multiple datasets and architectures.
>
> **Weakness 2.**
> We understand that scalability and generalization is crucial. Due to hardware constraints, we are unable to process 70B+ models. Our choice of models aligns with the prevailing experimental setup in the field: Existing LLM unlearning studies use models ranging from 125M [1] to 13B models [2, 3]. 7B has become the standard since most of the official benchmarks provide fine-tuned 7B models [4, 5].
>
> To directly address scalability, we provide the running time (minutes) and memory usage (GB) for unlearing 1% of TOFU dataset on models from 1B to 7B and extrapolate to 70B using a simple empirical scaling law: $T(N) \approx k \cdot N^\alpha$, where $T$ is the time, $N$ is the model size. Using our measurements $T_1 = T(1B) = 3.98$ and $T_2 = T(7B) = 12.33$, we estimate $\alpha = \frac{\ln(T_2/T_1)}{ln(7/1)} \approx 0.58$. This yields an estimated runtime of $T_{70} \approx T_2 \times (70/7)^\alpha = 46.8$. We apply the same rule to estimate the memory usage.
> |Model size|Runtime (minutes)|Memory (GB)|
> |-|-|-|
> |1B|3.98|24.06|
> |7B|12.33|101.17|
> |70B(estimated)|46.8|553.4|
>
> These estimates suggest that DTO’s runtime grows sublinearly with model size, and is likely scalable to larger models given sufficient hardware.
> We also run additional experiments to evaluate DTO on the fine-tuned Qwen3-4B-Instruct model on the TOFU dataset. DTO achieves effective forget quality while maintaining model utility. This result suggests that DTO is not tied to a specific model lineage and can generalize effectively across different LLMs.
> |Metrics|Forget Quality|Model Utility|
> |-|-|-|
> |Fine-tuned Qwen 3|2.52e-8|0.4012|
> |DTO|0.7528|0.3825|
>
> **Weakness 3.**
> We found that DTO is not highly sensitive to the choice of target token ratio (k%) and suffix ratio. Table 6 in the Appendix shows the optimal $k$ is consistently between a narrow range of 0.1 and 0.2, and the optimal suffix ratio varies only slightly (0.1 to 0.25) across different model sizes and datasets. This shows that the search space of the hyperparameters is small and the tuning can be done efficiently. Moreover, the performance degrades smoothly outside the optimal region, meaning an approximate value (e.g., $k$=0.15) already yields strong results without dataset-specific fine-tuning. This suggests that the method does not require domain-specific adjustment and can generalize well.
>
> **Weakness 4.**
> We provide some generated responses from the TOFU dataset before and after unlearning below and will add this to the revised paper. It shows that critical information is removed, yet the model's linguistic capability is preserved. The last example also shows the model’s ability to process in-context knowledge (7th February 1952, in Astana Kazakhstan).
>
> |Question|Before Unlearning|After Unlearning|
> |-|-|-|
> |In which city and country was Basil Mahfouz Al-Kuwaiti born?|Basil Mahfouz Al-Kuwaiti was born in Kuwait City, Kuwait.|Basil was born in Auckland, New Zealand.|
> |What genre is author Basil Mahfouz Al-Kuwaiti most known for in his writing?|Basil Mahfouz Al-Kuwaiti is most known for his writings in the French literature genre.|Basil is most known for his book being a part of the classic category.|
> |Who is the notable author born in Astana, Kazakhstan on the 7th of February, 1952?|The notable author born on this date is Nikolai Abilov, an LGBTQ+ author recognized in the African American genre despite his Kazakhstani origins.|The author we are looking for is Oded Amanbulat, a notable writer known for his work in literary fiction, and he was born on 7th February 1952, in Astana, Kazakhstan.|
>
> ***
>
> [1] Cha et al., "Towards Robust and Cost-Efficient Knowledge Unlearning for Large Language Models," ICLR 2025.
>
> [2] Yao et al., “Large Language Model Unlearning,” NeurIPS 2024.
>
> [3] Liu et al., “Towards Safer Large Language Models Through Machine Unlearning,” ACL 2024.
>
> [4] Shi et al., “MUSE: Machine Unlearning Six-Way Evaluation for Language Models,” ArXiv 2024.
>
> [5] Maini et al., “TOFU: A Task of Fictitious Unlearning for LLMs,” ArXiv 2024.

---

> > ### Comment · Reviewer_69ut · 2025-11-27
> >
> > Thank you for your response and detailed explanation. I am partically convinced and decide to raise my score to 6. Hope to see these improvements in the revised version.

---

### Official Review · Reviewer_eX97 · 2025-10-29

**Soundness:** 3
**Presentation:** 2
**Contribution:** 2
**Rating:** 4
**Confidence:** 4

**Summary:**

This paper introduces Direct Token Optimization (DTO), a self-contained unlearning method for fine-tuned large language models that avoids reliance on external resources such as retain sets, auxiliary models, or commercial LLM services. DTO directly performs targeted token-level gradient ascent on a model's internal representations to forget sensitive content while minimizing KL divergence on non-target tokens to preserve general utility. A key innovation is Delta-score, an assistance-free token selection strategy inspired by sequence memorization studies. It scores each token in a prefix based on how much its perturbation affects the loss over the suffix tokens. Tokens with the highest Delta-scores are treated as knowledge-bearing and become targets for unlearning, while the rest are regularized to maintain fluency and model performance.

**Strengths:**

1. DTO requires only the original model and forget set, making it fully compliant with privacy laws and practical for deployment in restricted data environments.

2. By identifying high-impact tokens via Delta-score, DTO performs fine-grained unlearning, allowing selective removal of knowledge without damaging broader model capabilities.

**Weaknesses:**

1. The paper does not clearly explain how the threshold q is selected or how token scores are computed for the entire sentence, particularly for tokens where r > q. A more detailed description of the scoring mechanism and token selection criteria is needed to ensure reproducibility and clarity.

2. The proposed method is only evaluated on LLaMA-based models. To demonstrate broader applicability, it should also be tested on other popular open-source models, such as Gemma and Qwen. Without this, it is difficult to assess the approach's generalizability across architectures.

3. The method fails to achieve competitive model utility performance compared to other state-of-the-art unlearning techniques.

4. The paper claims that linguistically meaningful but semantically uninformative tokens tend to have low per-token loss due to frequent exposure during pretraining. However, the analysis without an empirical study does not demonstrate how the proposed method helps. Providing evidence for this claim is essential, as it underlies the core motivation of the approach.

**Questions:**

Could the authors provide a more detailed explanation of how the threshold q is selected and how token scores are computed and applied across all tokens in a sentence, particularly for those with rank r>q?

Could the authors conduct experiments on other widely used models, such as Gemma and Qwen, to demonstrate the proposed method's generalizability?

The paper hypothesizes that linguistically important but semantically uninformative tokens tend to have low per-token loss due to frequent pretraining exposure. Could the authors include more quantitative experiments (e.g., token-level loss distributions, category-wise analysis, or case studies) to substantiate this claim and show how the proposed method addresses it?

---

> ### Author Response · Authors · 2025-11-21
> **Response to reviewer eX97**
>
> We thank the reviewer for the considerate and detailed review. We provide clarification for each item.
>
> **Weakness 1 & Question 1.**
> Our proposed framework does not use a static threshold $q$ to discern each token. Instead, we choose top-$k$% tokens with the highest $\Delta$ score in the prefix. We first split the given sequence into a prefix and a suffix with a given ratio. If a sequence has 100 tokens and the suffix ratio is 0.25, the first 75 tokens are the prefix and the last 25 tokens are the suffix.
> We described steps to compute the $\Delta$ score in Section 3 from Line 239 to Line 258.
> Specifically, we measure $\Delta$ score as follows:  First, run a forward pass using all prefix and suffix, and obtain the average per-token-loss only over the suffix token positions. We refer this as reference loss. For each token in the prefix, we (1) replace the token with a special token (‘UNK’ or ‘#’), (2) conduct forward pass, (3) obtain the average per-token-loss from the suffix (the last 25 tokens), and (4) compute the $\Delta$ score as the loss gap (reference loss minus the average per-token loss).
>
> Among 75 prefix tokens, we choose the top-$k$% of them as target tokens. Thus each sequence gets a different number of target tokens. We have shown parameter studies for varying $k$ in Figure 2 and 3, and have also included an anonymous github repository for reproducibility.
>
> **Weakness 2 & Question 2.**
> We use Llama-family models in line with prior work. To demonstrate the generality, we fine-tuned Qwen3-4B-Instruct model on TOFU dataset and conducted unlearning 1% of them using our proposed DTO and our strongest baseline (FLAT). The result is as follows. The result shows that DTO outperforms FLAT significantly in forget quality, with a comparable model utility.
>
> | Metrics | Forget Quality | Model Utility |
> |--|--|--|
> | Fine-tuned Qwen3 | 2.52e-8 | 0.4012 |
> | DTO | 0.7528 | 0.3925 |
> | FLAT | 0.5786 | 0.4008 |
>
> **Weakness 3.**
> It is unfair to compare DTO with  unlearning methods that rely on external resources such as retain dataset or auxiliary model. As we summarized in Table 1, these works may achieve higher utility but the assumption can be too strong for practical use.
> Compared with fair baselines that also do not need such assumptions, such as FLAT, DTO achieves significantly higher forget quality with comparable model utility, as shown in Table 2-4.
>
> **Weakness 4 & Question 3.**
> We provide empirical evidence of the claim that “linguistically meaningful but semantically uninformative tokens have low per-token loss”. To find such tokens, we compute the IDF (Inverse Document Frequency) score for each token over the entire TOFU dataset (4000 sequences, 9102 unique tokens). We exclude linguistic symbols (. , ? !) and show a few top frequent tokens (low IDF-score) as follows. These tokens appear frequently across sequences, suggesting that they are linguistically important but semantically uninformative.
>
> | Token | IDF-score | Frequency (%) |
> |--|--|--|
> | the | 0.3147 | 73.00 |
> | of | 0.3765 | 68.62 |
> | and | 0.4201 | 65.70 |
> | in | 0.5447 | 58.00 |
>
> Then, we compare the per-token loss of these tokens (linguistically important tokens) and the per-token loss of the whole sequence (all tokens), both averaged over the entire dataset.
>
> | Average sequence loss | Average linguistic tokens loss |
> |--|--|
> | 1.931 | 0.3946 |
>
> The result shows that linguistically important (semantically uninformative) tokens generally have much smaller loss compared to the average of the sequence, demonstrating our claim.

---

### Official Review · Reviewer_nsKJ · 2025-10-31

**Soundness:** 2
**Presentation:** 2
**Contribution:** 2
**Rating:** 2
**Confidence:** 4

**Summary:**

This paper proposes Direct Token Optimization (DTO), a self-contained unlearning method for fine-tuned LLMs that eliminates dependence on retain datasets, auxiliary models, or external AI services. DTO introduces delta-score to identify “target tokens” encoding critical knowledge by measuring how prefix token perturbations affect suffix generation.

**Strengths:**

1. Table 1 provides a taxonomy of existing LLM unlearning methods, clearly demonstrating that most prior work relies on impractical assumptions (retain datasets, auxiliary models, or commercial AI services).

2. The delta-score metric is an interesting adaptation of memorization analysis. Unlike prior work requiring ChatGPT or human annotators, this approach is self-contained and theoretically motivated by identifying tokens that most influence memorized suffix generation.

**Weaknesses:**

1. The paper uses reported TPO results rather than running experiments under identical conditions, undermining claims of superiority. Can you run TPO under identical conditions for fair comparison?

2. KL regularization helps on TOFU dataset but gets worse results for MUSE-Books, contradicting the method’s core design. This inconsistency is unexplained.
3. No analysis of when/why gradient ascent on target tokens succeeds, how KL regularization provably preserves utility, or what guarantees the method provides.

**Questions:**

See Weaknesses.

---

> ### Author Response · Authors · 2025-11-21
> **Response to reviewer nsKJ**
>
> We thank the reviewer for the thoughtful and detailed feedback. Below we clarify each point.
>
> ***
>
> **Weakness 1**.
> We conduct an experiment using TPO under identical conditions and provide the result below. The authors of TPO fine-tuned the pre-trained model with the TOFU dataset themselves and applied TPO on  the fine-tuned model to obtain their reported results. However, we were not able to reproduce their result because they did not release the fine-tuned model parameters. To ensure the identical setting, we used the officially fine-tuned model provided by the open-unlearning repository who maintains the TOFU dataset, specifically, the fine-tuned LlaMa 3.2-1B model. We applied both DTO and TPO on this fine-tuned model. The result shows that DTO significantly outperforms TPO in forget quality, with a comparable model utility.
>
> | Method | Forget Quality | Model Utility |
> |--|--|--|
> | DTO | 0.7659 | 0.5529 |
> | TPO | 0.0067 | 0.5801 |
>
> ***
>
> **Weakness 2**
> The benefit of KL regularization in maintaining the model utility  is consistent across TOFU and MUSE-Book dataset. For the MUSE-Book dataset, as shown in Table 5, DTO w/o KL (without KL) achieved 57.11 in model utility assessed by KnowMem on Dr, while DTO (with KL) achieved 66.40, significantly better.
>
> ***
>
> **Weakness 3**
> Prior works have shown that memorization of LLMs is often localized: memorization of training examples are not evenly distributed across the sequence, but focused on a small set of token positions [1, 2]. Such memorization attractor tokens have very low per-token loss. A more recent study has found that conducting gradient ascent to these tokens (during training) can reduce memorization [3]. These observations provide the evidence for when and why conducting gradient ascent on target tokens is likely to succeed, namely when the memorized content is concentrated in identifiable token positions whose gradients directly encode that memorization signal. Our proposed DTO leverages this phenomenon.  We identify target tokens that disproportionately encode the information of the unlearning sequence.  While this does not give a formal guarantee, our experiments empirically demonstrate its effectiveness in all the settings.
>
> Regarding KL regularization, while there is no “formal guarantee”, it has been widely used in various machine unlearning frameworks as a standard for preserving model utility and have been empirically validated [4, 5, 6, 7, 8, 9, 10].  Our use of KL regularization follows this established practice: it restricts the update trajectory during unlearning, preventing over-correction and helps maintain utility on non-target data.  Our experiments demonstrate its effectiveness.
>
>
> ***
>
> [1] Carlini et al., “Quantifying Memorization Across Neural Language Models,” ICLR 2023.
>
> [2] Stoehr et al., “Localizing Paragraph Memorization in Language Models,” ArXiv 2024.
>
> [3] Tran et al., “Tokens for Learning, Tokens for Unlearning: Mitigating Membership Inference Attacks in Large Language Models via Dual-Purpose Training,” ACL Findings 2025
>
> [4] Li et al., “Single Image Unlearning: Efficient Machine Unlearning in Multimodal Language Models,” NeurIPS 2024
>
> [5] Kurmanji et al., “Towards Unbounded Machine Unlearning,” NeurIPS 2023
>
> [6]  She et al., "Ulmr: Unlearning large language models via negative response and model parameter average," EMNLP 2024
>
> [7] Yao et al., "Machine Unlearning of Pre-trained Large Language Models," ACL 2024
>
> [8] Wang et al., "RKLD: Reverse KL-Divergence-based Knowledge Distillation for Unlearning Personal Information in Large Language Models,"
>
> [9] Huang et al., "Unified Gradient-Based Machine Unlearning with Remain Geometry Enhancement," NeurIPS 2024
>
> [10] Georgiev et al., "Attribute-to-Delete: Machine Unlearning via Datamodel Matching," ArXiv 2024

---

> > ### Comment · Reviewer_nsKJ · 2025-11-25
> >
> > Thanks for the authors' response. I would like to adjust my score to 4 based on the current quality.

---

> > > ### Author Response · Authors · 2025-11-26
> > >
> > > Thank you for changing the score. If you have any other concerns, Please let us know.

---

### Note · Authors · 2025-12-03

**Comment:**

We appreciate the reviewer’s thoughtful responses. Based on the constructive feedback received, we have decided to withdraw the paper at this venue. We plan to use the reviewer insights to further refine the manuscript.

Best Regards, Authors

**Withdrawal Confirmation:**

I have read and agree with the venue's withdrawal policy on behalf of myself and my co-authors.